# Immediate and short-term effects of eccentric muscle contractions on structural, morphological, mechanical, functional and physiological properties of peripheral nerves: A protocol for a systematic review and meta-analysis

**Dorina Lungu**[1], **Tiago Neto**[2], **Ricardo J. Andrade**[3,4], **Michel W. Coppieters**[4,5], **Raúl Oliveira**[1,6], **Sandro R. Freitas**[1] *

**1** Neuromuscular Research Lab, Faculty of Human Kinetics, University of Lisbon, Cruz Quebrada, Lisbon, Portugal, **2** Luxembourg Institute of Research in Orthopedics, Sports Medicine and Science, Luxembourg, Luxembourg, **3** Laboratory «Movement, Interactions, Performance» (EA 4334), Faculty of Sport Sciences, Nantes, University of Nantes, Nantes, France, **4** Menzies Health Institute Queensland, Griffith University, Brisbane and Gold Coast, Queensland, Australia, **5** Faculty of Behavioural and Movement Sciences, Vrije Universiteit Amsterdam, Amsterdam Movement Sciences, Amsterdam, The Netherlands, **6** Interdisciplinary Centre for the Study of Human Performance, Faculty of Human Kinetics, University of Lisbon, Cruz Quebrada, Lisbon, Portugal

* sfreitas@fmh.ulisboa.pt

## Abstract

### Background

It is widely acknowledged that eccentric muscle contractions may cause skeletal muscle damage. However, there is little knowledge about the impact of eccentric contractions on non-muscular structures. Animal and human studies revealed that eccentric contractions can also induce immediate and short-term nerve dysfunction. The purpose of this review is to examine whether eccentric muscle contractions induce immediate and short-term effects on structural, morphological, mechanical, functional and physiological properties of peripheral nerves, from both animal and human studies.

### Methods and analysis

A systematic review of randomised (RCTs) and non-randomised controlled trials will be conducted. Four electronic databases (i.e., Medline/Pubmed, Science Direct, PEDro and Cochrane) will be searched using predefined search terms to identify relevant studies. Eligible studies have to comprise any type of eccentric contraction of upper or lower limb muscles. Primary outcomes will include measures related to structure, morphology, mechanical, functional and physiological properties of peripheral nerves. Two independent reviewers will assess eligibility, evaluate risk of bias, and extract relevant data. In human studies, the risk of bias will be assessed by the Cochrane Collaboration risk of bias tool (RoB 2.0 tool) for

**Data Availability Statement:** No datasets were generated or analysed during the current study. All relevant data from this study will be made available upon study completion.

**Funding:** The research center CIPER – Centro Interdisciplinar para o Estudo da Performance Humana (UID/00447/2020), supported by Fundação para a Ciência e a Tecnologia and hosted by Faculdade de Motricidade Humana – Universidade de Lisboa, provides financial support to Raúl Alexandre Nunes da Silva Oliveira, regarding the payment of the publication fee of the manuscript "Immediate and short-term effects of eccentric muscle contractions on structural, morphological, mechanical, functional and physiological properties of peripheral nerves: a protocol for a systematic review and meta-analysis", for the journal PLOS One. Raúl Oliveira was supported by Fundação para a Ciência e a Tecnologia, under grant number UID/00477/2020 awarded to CIPER - Centro Interdisciplinar para o Estudo da Performance Humana (unit 447).

**Competing interests:** The authors have declared that no competing interests exist.

**Abbreviations:** BDNF, Brain-derived neurotrophic factor; CK, Creatine kinase; CSA, Cross sectional area; GAP-43, Growth-associated protein 43; MAG, Myelin-associated glycoprotein; MRI, Magnetic resonance imaging; NCV, Nerve Conduction Velocity; NGF, Nerve growth factor; NT-3, Neurotrophin 3; NT-4/5, Neurotrophin 4/5; p0, Myelin sheath protein zero; PMP22, Peripheral myelin protein 22; PRISMA-P, Preferred Reporting Items for Systematic Reviews and Meta-Analyses Protocols; PROSPERO, Prospective Register of Systematic Reviews; RCTs, Randomised controlled trials; RoB 2.0 tool, Cochrane Collaboration risk of bias tool; ROBINS-I, Risk Of Bias In Non-randomised Studies—of Interventions; TrkC, Tropomyosin receptor kinase C; US, Ultrasound.

RCTs and by risk of bias in non-randomised studies of interventions (ROBINS-I) for non-randomised controlled trials; while for animal studies, the risk of bias will be assessed using the SYRCLE's RoB tool. A narrative synthesis will be conducted for all included studies. Also, if appropriate, a meta-analysis will be performed, where the effect size of each outcome will be determined by the standardized mean difference as well as the 95% confidence intervals. $I^2$ statistics will be used to assess heterogeneity.

## Ethics and dissemination

For this study, no ethical approval is required. Findings will be disseminated widely through peer-reviewed publication and conference presentations.

## Systematic review registration

The protocol has been registered at the International Prospective Register of Systematic Reviews (PROSPERO). Registration number: CRD42021285767.

## 1. Introduction

Eccentric contractions of skeletal muscles are contractions where the muscle-tendon unit is lengthened while the muscle is activated [1]. There is a considerable body of literature showing that eccentric contractions may induce signs and symptoms of muscular damage [2]; in particular to unaccustomed eccentric contractions [3], performed at high intensity [4] and with fast muscle-tendon elongation velocity [5]. Manifestations of muscle damage include delayed onset muscle soreness [6], functional neuromuscular deficits as prolonged strength loss [7], decreased joint range of motion [7], muscle swelling [8], increased muscular stiffness [9], and increase of muscle proteins in the blood (e.g., creatine kinase, lactate dehydrogenase, and myoglobin) [10]. However, literature is scarce concerning the functional and mechanical impact of eccentric contractions on non-muscular structures, such as peripheral nerves.

Skeletal muscle and peripheral nerves are known to be anatomically linked, not only through the neuromuscular junction but also via neuromyofascial structures [11]. Due to the anatomical connectivity, muscle-nerve mechanical interactions occur during repetitive muscular contractions which leads to changes in nerve properties [12, 13]. These changes could potentially explain the loss of strength observed after eccentric or unaccustomed exercise. Accordingly, delayed sensory and motor conduction velocity of the radial nerve was observed at rest in the dominant arms of tennis players compared with their non-dominant arms and with control individuals [12]. Similarly, delays in distal and proximal M-wave latencies and decreased sciatic nerve conduction velocity (NCV) were observed in the injured limb of athletes who had experienced muscle strain injuries compared to uninjured side [13]. Taken together, these findings suggest reduced nerve function induced by mechanical stress, due to repetitive movements, muscle overload and muscle injury. However, recent investigations have focused on the acute effects of eccentric contractions on peripheral nerve properties. Several authors have shown acute changes in structural and functional properties of peripheral nerves in both animals [14, 15] and humans [16–19]. Thus, animal studies revealed myelin sheath damage, observed after 20 eccentric contractions of the plantar flexors with fast angular velocity (180°/s) [14]. The damage to the myelin sheath caused transient sciatic nerve dysfunction, manifested by a significant decline (i.e., -21%) in nerve conduction velocity (NCV)

observed on day 7 after eccentric contractions, which returned to baseline values on day 10 [14]. Also, temporary strength loss was present which gradually recovered over time; suggesting that a potential link may exist between the effects of nerve properties and the strength loss. Similar results were obtained after 4 repeated bouts of eccentric contractions at fast angular velocities (180˚/s) [15]. However, due to a higher volume of muscular contractions, a greater decrease of sciatic NCV (i.e., -42%) was observed, suggesting greater nerve damage. Concomitantly, decreased myelin thickness and nerve fiber diameter was observed [15]. These changes were accompanied by progressive decrease in plantar flexors torque across the bouts. Therefore, consecutive bouts of eccentric contractions with fast angular velocity cause severe nerve properties alterations, and thus impairing skeletal muscle function. Consistent with these animal studies findings, it has been shown that eccentric contractions induce a temporary reduction on nerve function in healthy individuals [16–19]. Therefore, delays in motor and sensory NCV of the median nerve were observed following 100 eccentric contractions of the flexor pollicis brevis muscle [16, 17]. Furthermore, an acute reduction (i.e., -27%) in muscle fiber conduction velocity was observed after 50 repeated maximal eccentric contractions of the elbow flexors, at 2 hours post-exercise [14]. It can be suggested that gross sarcolemmal function was impaired after eccentric exercise. This impairment of motor nerve function is present also at one and two days after eccentric contractions, as an increase on the M-wave latency was found, by 12% and 24%, respectively [19]. In summary, high intensity eccentric contractions may result in nerve disorders associated with disruption in neuromuscular junctions with concomitant impairment in action potential transmission and neuromuscular response.

Given the above, although the existing evidence about the mechanical interplay between the skeletal muscles and peripheral nerves during eccentric contractions, no systematic review analyzed the effects of eccentric contractions on peripheral nerves properties. To this end, this study aims to systematically review the literature in order to examine the immediate (i.e., <2 hours) and short-term (i.e., <10 days) effects of eccentric muscle contractions on peripheral nerve structural, morphological, mechanical, functional and physiological properties, from both animal and human studies. We will only consider findings from healthy individuals and animals without muscle or nerve pathology.

## 2. Methods

This protocol has been written according to the Preferred Reporting Items for Systematic Reviews and Meta-Analyses Protocols (PRISMA-P) guidelines [20] and has been registered on 17 November 2021 at the International Prospective Register of Systematic Reviews (PROSPERO) (Registration number: CRD42021285767).

### 2.1 Study eligibility criteria

A summary of the participants, interventions, comparators, and outcomes, as well as the type of studies included according to PICO strategy is shown in Table 1.

**2.1.1 Type of studies.** The review will include randomised (RCTs) and non-randomised controlled trials or quasi-experimental studies evaluating the effects of eccentric contractions (applied to upper or lower limb muscles) on peripheral nerves structural, morphological, mechanical, functional and physiological properties. Studies will be excluded if they are case reports, reviews, editorials, protocol studies and clinical guidelines. There will be no restrictions based on methodological quality and year of publication. Only the articles with the title and abstract published in English will be included for eligibility assessment.

**2.1.2 Types of participants.** This review will consider all studies that involve healthy asymptomatic individuals older than 18 years (irrespective of gender, race, ethnicity or other

**Table 1. A summary of the participants, interventions, comparators and outcomes considered, as well as the type of studies included according to PICO strategy.**

| PICO | Inclusion Criteria | Exclusion Criteria |
|---|---|---|
| **P**opulation | Human or animal studies. | Studies of animals or individuals presenting any type of pathology or age-related degenerative changes, such as neurodegenerative diseases, neuropathies and nerve transection. |
| **I**ntervention | Eccentric contractions of upper or lower limb muscles. | |
| **C**omparison | No other modality or different protocols of eccentric contractions (i.e., different number of series and sets, angular velocities, multiple bouts). | |
| **O**utcomes | **Structure and morphology** (from micro to macro levels): nerve fiber diameter and myelin sheath thickness (via microscopic analysis); fascicle number (i.e., axon density) and size (i.e., diameter) by high frequency ultrasonography or magnetic resonance microscopy; nerve thickness by ultrasound (US) or magnetic resonance imaging (MRI) or microscopic analysis; nerve cross sectional area (CSA) by US or MRI or microscopic analysis; nerve volume with freehand 3-D US or MRI. <br> **Mechanics:** nerve strain; excursion, and stiffness via US and US elastography (shear wave elastography). <br> **Function:** sensory and motor nerve function—nerve conduction velocity (NCV). <br> **Physiology:** intraneural blood flow measured with Doppler US (B-mode with Colour Doppler) or laser doppler flowmetry; change in molecular expressions (proteins): myelin sheath protein zero (p0); growth-associated protein 43 (GAP-43); myelin-associated glycoprotein (MAG); peripheral myelin protein 22 (PMP22); galectin-3/MAC-2; *tropomyosin receptor kinase C (*TrkC); neurotrophins [nerve growth factor (NGF), brain-derived neurotrophic factor (BDNF), neurotrophin 3 (NT-3) and neurotrophin 4/5 (NT-4/5)] (through immunohistochemistry); serum levels of neuroinflammatory mediators (or through immunohistochemistry) such as cytokines, neuropeptides, reactive oxygen species, and chemokines; alterations in signal intensity (T1 and T2 via MRI) and echo intensity (via high-resolution US). | Studies that do not report any variable of nerve damage or dysfunction of interest. |
| Study design | Randomised controlled trial and non-randomised controlled trials (quasi-experimental studies). | Non-primary literature, such as reviews, editorials, protocol studies, clinical guidelines, and case reports. |

demographic characteristics) and animal (regardless species, sex, weight and age) studies. Reports including individuals and animals presenting any type of pathology or age-related degenerative changes, such as neuropathies, neurodegenerative diseases and nerve transection, will be excluded.

**2.1.3 Type of intervention.** Eligible studies must include any type of eccentric contractions of upper or lower limb muscles.

**2.1.4 Types of outcome measures.** We will evaluate the immediate (i.e., up to 2 hours after eccentric contraction) and short term (i.e., up to 10 days) effects of eccentric contractions on structural, morphological, mechanical, functional and physiological properties of the peripheral nerves. Studies with at least one of following outcomes will be included:

1. **Structure and morphology** (from micro to macro levels): nerve fiber diameter and myelin sheath thickness (via microscopic analysis); fascicle number (i.e., axon density) and size (i.e., diameter) by high frequency ultrasonography or magnetic resonance microscopy; nerve thickness by ultrasound (US) or magnetic resonance imaging (MRI) or microscopic analysis; nerve cross sectional area (CSA) by US or MRI or microscopic analysis; nerve volume with freehand 3-D US or MRI.

2. **Mechanics:** nerve strain; excursion, and stiffness via US and US elastography (shear wave elastography).

3. **Function:** sensory and motor nerve conduction velocity (NCV).

4. **Physiology:** intraneural blood flow measured with Doppler US (B-mode with Colour Doppler) or laser doppler flowmetry; change in molecular expressions (proteins): myelin sheath protein zero (p0); growth-associated protein 43 (GAP-43); myelin-associated glyco-protein (MAG); peripheral myelin protein 22 (PMP22); galectin-3/MAC-2; tropomyosin receptor kinase C (TrkC); neurotrophins [nerve growth factor (NGF), brain-derived neuro-trophic factor (BDNF), neurotrophin 3 (NT-3) and neurotrophin 4/5 (NT-4/5)] (through immunohistochemistry); serum levels of neuroinflammatory mediators (or through immu-nohistochemistry) such as cytokines, neuropeptides, reactive oxygen species, and chemo-kines; alterations in signal intensity (T1 and T2 via MRI) and echo intensity (via high-resolution US).

## 2.2. Search methods for identification of studies

**2.2.1 Search strategy.** This literature review will be based on systematic searches in four literature electronic databases, including Medline/Pubmed, Science Direct, PEDro and Cochrane. All search terms and their combination using the Boolean operators 'AND' and 'OR' are included in Table 2.

*2.2.1.1 Additional search methods (grey literature).* To identify studies not captured by our electronic databases search, we will hand-search and screen reference lists of relevant articles and of included studies to try to identify other potentially eligible trials or ancillary publications.

## 2.3 Data collection and analysis

**2.3.1 Study selection.** Initial screening of studies will be based on title and abstract (after removing duplicate records) and will be conducted independently by two authors (DL and TN). The full text of potentially relevant studies will be retrieved, and the full text of the articles

**Table 2. Search terms and their combination.**

| Search terms and their combination |
| --- |
| 1. (Eccentric OR "active lengthening" OR "lengthening muscle contractions" OR "lengthening contractions" OR "lengthening action" OR "muscle lengthening actions" OR "negative work") AND (nerve OR "peripheral nerve" OR "nerve damage" OR "nerve damages" OR "nerve damaging" OR "nerve impairment" OR "nerve dysfunction" OR "nerve disorder" OR "nerve injury" OR "motor unit injury" OR "motor unit damage") |
| 2. 1 AND ("nerve morphology" OR "nerve structure" OR "nerve microstructure" OR "nerve macrostructure" OR "nerve fascicle" OR "nerve fasciculus" OR "nerve fiber" OR "nerve fiber diameter" OR "nerve fascicular size" OR "fascicle number" OR "axon" OR "axon diameter" OR "axon density"OR "myelin sheath" OR "myelin sheath thickness" OR "nerve sheath" OR "nerve sheath diameter" OR "Schwann cells" OR "nerve cell" OR "neuron" OR "neurons" OR "connective tissue" OR "connective tissue layers" OR endoneurium OR perineurium OR epineurium) |
| 3. 1 AND ("nerve mechanics" OR neuromechanics OR "nerve mechanical properties" OR "nerve mechanical changes" OR "nerve biomechanics" OR "nerve biomechanical properties" OR "nerve strain" OR "nerve stiffness" OR "shear wave elastography" OR "shear wave velocity" OR "nerve excursion" OR "nerve gliding" OR "nerve sliding" OR "nerve displacement" OR "nerve mobility" OR ultrasound OR elastography OR "shear wave elastography") |
| 4. 1 AND ("nerve function" OR "neuromuscular junction" OR "nerve functional changes" OR "Nerve conduction velocity" OR NCV OR "neural conduction" OR "motor nerve conduction velocity" OR "sensory nerve conduction velocity" OR "sensory nerve action potential" OR SNAP OR "muscle fiber conduction velocity" OR "compound muscle action potential" OR CMAP OR "motor unit conduction velocity" OR "motor nerve impairment" OR "action potential conduction" OR "action potential transmission" OR "action potential propagation velocity" OR "nerve impulse" OR "action potential propagation velocity" OR "sarcolemmal action potential conduction" OR "neurophysiological properties") |
| 5. 1 AND ("nerve physiological properties" OR "nerve physiology" OR "blood flow" OR "blood perfusion" OR "intraneural blood flow" OR "blood flow velocity" OR "Functional MRI" OR "Doppler ultrasound" OR "laser doppler flowmetry" OR "change in molecular expressions" OR "myelin sheath protein zero" OR p0 OR "Growth-associated protein 43" OR GAP-43 OR "Tropomyosin receptor kinase C" OR TrkC OR "myelin-associated glycoprotein" OR MAG OR "peripheral myelin protein 22" OR PMP22 OR" galectin-3" OR MAC-2 OR neurotrophins OR "nerve growth factor" OR NGF OR "brain-derived neurotrophic factor" OR BDNF OR "neurotrophin 3" OR NT-3 OR neurotrophin 4/5 OR NT-4/5 OR "nerve thickness" OR "nerve cross sectional area" OR CSA OR volume OR ultrasound OR "freehand 3-D ultrasound" OR "magnetic resonance imaging"OR MRI OR "high frequency ultrasonography" OR "magnetic resonance microscopy" OR "neuroinflammatory mediators" OR "inflammatory mediators" OR cytokines OR neuropeptides OR "neurotrophic factors" OR "reactive oxygen species" OR chemokines) |
| 6. 1 AND ("nerve echogenicity" OR "peripheral nerve imaging" OR "signal intensity" OR "Magnetic Resonance Imaging" OR MRI OR "Magnetic Resonance Neurography" OR "Functional MRI" OR "Diffusion Tensor Imaging" OR DTI OR microneurography OR Ultrasonography OR ultrasound OR "B-mode ultrasonography" OR "ultrasound echo intensity" OR "echo intensity" OR echostructure OR sonoelastography OR "Shear-wave sonoelastography") |

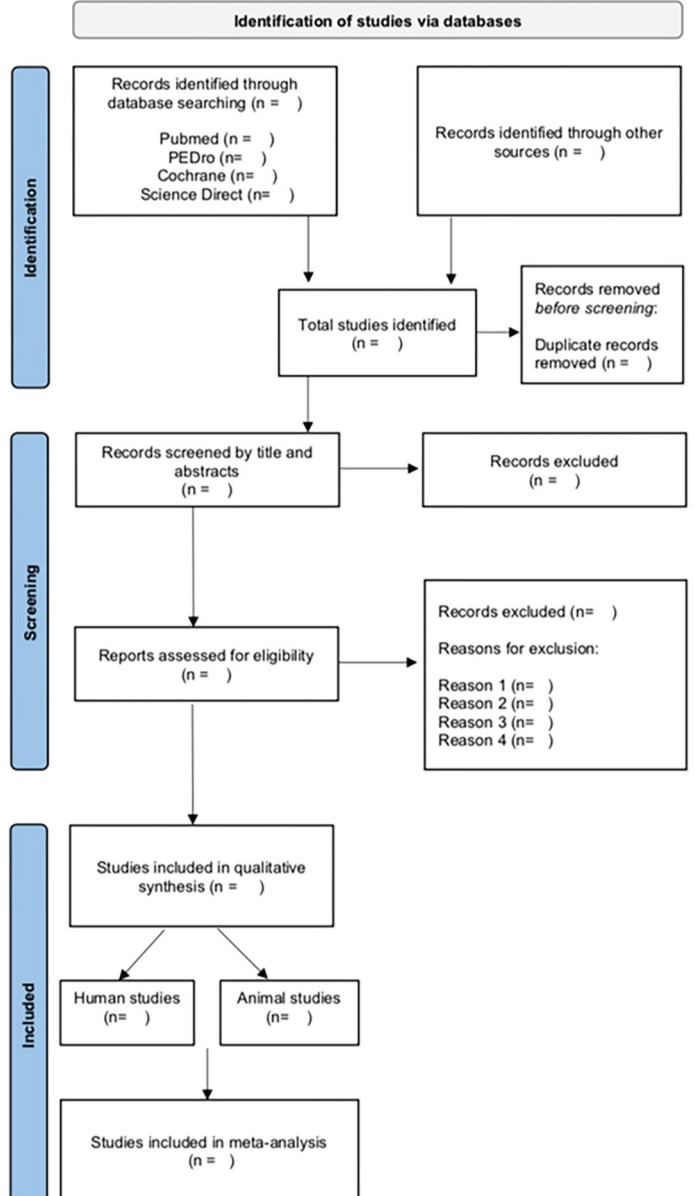

**Fig 1. The PRISMA flow diagram of study selection process.**

will be screened independently for eligibility by two authors (DL and TN). Reasons for exclusion will be recorded. In case of disagreement, the two reviewers will attempt to resolve any discrepancies. If the disagreement persists, a third reviewer (RA, MC, RO or SF) will settle the disagreement and will make a final decision. The PRISMA flow diagram provides an overview of the study selection process (Fig 1).

**2.3.2 Data extraction.** Data extraction will be performed independently by two authors (DL and TN). The following information will be extracted from each study and summarized in spreadsheets: (1) identification of the study (i.e., article title; authors; publication year); (2) methodological characteristics (i.e., study design; study objective or research question or hypothesis; sample characteristics (i.e., sample size; sex; age; human/animals); groups and controls: characteristics of eccentric contractions (e.g., number of repetitions; number of series;

angular velocity; number of sessions, muscle (group) targeted; type of control condition; outcomes measured (e.g., change in molecular expressions; nerve fiber diameter; myelin sheath thickness; fascicle number and size; nerve thickness; nerve CSA; nerve volume; nerve strain; nerve excursion; nerve stiffness; intraneural blood flow; NCV); stated length of follow-up (after intervention); validated measures; statistical analyses] and (3) main findings. If the outcome data in the original article were unclear or missing, the corresponding author will be contacted via email for clarification.

### 2.4 Risk of bias assessment

Methodological quality of the RCTs will be assessed using the Cochrane Collaboration risk of bias tool (RoB 2.0 tool), the most recommended tool for RCTs [21]. The RoB 2.0 tool consists of six domains of bias: selection bias, performance bias, detection bias, attrition bias, reporting bias, and any other biases. According to specific criteria available in the Cochrane Collaboration risk of bias tool, the two reviewers will categorize the bias as "low risk of bias," "unclear risk of bias," and "high risk of bias" [22].

For assessing non-randomised interventional studies, the ROBINS-I will be used. This tool is guided through seven chronologically arranged bias domains (pre-intervention, at intervention, and post-intervention), and the interpretations of domain-level and overall risk of bias judgement in ROBINS-I are classified in low, moderate, serious, or critical risk of bias [23].

Animal studies will be assessed using the RoB tool for animal intervention studies–SYRCLE's RoB tool, based on the original Cochrane RoB Tool, which had become the most recommended tool for assessing the methodological quality of animal intervention studies [24]. This tool contains 10 items related to 6 types of bias: selection bias, performance bias, detection bias, attrition bias, reporting bias and other biases. In order to assign a judgment of low, high or unclear risk of bias to each item mentioned in the tool, it is proposed a detailed list with signaling questions to aid the judgment process. A "yes" score indicates low risk of bias while "no" indicates high risk of bias. The score will be "unclear" if insufficient details have been reported to assess the risk of bias properly [24].

Two independent reviewers (DL and TN) will assess the risk of bias of eligible trials of this systematic review. Reviewers will score the selected studies and the results will be presented on a table with an explicit judgment of quality of evidence. Disagreements will be resolved by a consensus meeting between both reviewers or by a third reviewer (RA, MC, RO or SF). Inter-rater agreement will be evaluated using Kappa statistics.

### 2.5 Data synthesis

If studies are sufficiently homogeneous in terms of design and comparator, results of comparable groups of studies will be pooled into statistical meta-analysis. Meta-analyses for the animal and human studies will be performed separately and applied for each outcome of interest of nerve properties. To measure the strength and relationship between variables, the effects of eccentric contractions on nerve structure, morphology, mechanical, functional and physiological properties, will be presented using effect sizes. Effect sizes will be determined by the standardized mean difference and 95% confidence intervals will be calculated. Heterogeneity will be determined by conducting a $I^2$ test. If statistical assessment will not be possible due to low number of studies for each outcome, the findings will be presented in narrative form.

## 3. Discussion

This systematic review will provide a detailed summary of the evidence for the effects of eccentric contractions on peripheral nerve properties. From the preliminary search, alterations in

some outcomes of interest under study are expected, such as decreased nerve function, compatible with peripheral nerve damage, both from animals and human studies.

The review results will inform the design of a trial evaluating the effect of eccentric intervention on nerve properties.

## 4. Dissemination and knowledge transfer

Findings will be disseminated widely through peer-reviewed publication and in various media, for example, conferences, congresses, or symposia.

## Supporting information

**S1 Checklist. PRISMA-P (Preferred Reporting Items for Systematic review and Meta-Analysis Protocols) 2015 checklist: Recommended items to address in a systematic review protocol\*.**
(PDF)

## Author Contributions

**Conceptualization:** Dorina Lungu, Tiago Neto, Ricardo J. Andrade, Michel W. Coppieters, Raúl Oliveira, Sandro R. Freitas.

**Methodology:** Dorina Lungu, Tiago Neto, Ricardo J. Andrade, Michel W. Coppieters, Raúl Oliveira, Sandro R. Freitas.

**Writing – original draft:** Dorina Lungu.

**Writing – review & editing:** Dorina Lungu, Tiago Neto, Ricardo J. Andrade, Michel W. Coppieters, Raúl Oliveira, Sandro R. Freitas.

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
