## [Decision Letter · Decision Letter 0]

26 Apr 2023

PONE-D-22-32928

Immediate and short-term effects of eccentric muscle contractions on structural, morphological, mechanical, functional and physiological properties of peripheral nerves: a protocol for a systematic review and meta-analysis

PLOS ONE

Dear Dr. Lungu,

Thank you for submitting your manuscript to PLOS ONE. After careful consideration, we feel that it has merit but does not fully meet PLOS ONE’s publication criteria as it currently stands. Therefore, we invite you to submit a revised version of the manuscript that addresses the points raised during the review process.

The reviewers were positive about the manuscript and have requested minor changes that they perceive will add clarity to the study, as well as areas you may wish to consider including or enhancing. 

We look forward to receiving your revised manuscript.

Kind regards,

Charlie M. Waugh

Academic Editor

PLOS ONE

“No sources of funding were used to assist in the preparation of this systematic review

protocol.”

“The authors declare that they have no competing interests”

Reviewers' comments:

Reviewer's Responses to Questions

**Comments to the Author**

1. Does the manuscript provide a valid rationale for the proposed study, with clearly identified and justified research questions?

Reviewer #1: Yes

Reviewer #2: Yes

2. Is the protocol technically sound and planned in a manner that will lead to a meaningful outcome and allow testing the stated hypotheses?

Reviewer #1: Yes

Reviewer #2: Yes

3. Is the methodology feasible and described in sufficient detail to allow the work to be replicable?

Reviewer #1: Yes

Reviewer #2: Yes

4. Have the authors described where all data underlying the findings will be made available when the study is complete?

Reviewer #1: No

Reviewer #2: Yes

5. Is the manuscript presented in an intelligible fashion and written in standard English?

Reviewer #1: Yes

Reviewer #2: Yes

6. Review Comments to the Author

You may also provide optional suggestions and comments to authors that they might find helpful in planning their study.

Reviewer #1: Reviewer Comments to the manuscript PONE-D-22-32928, “Immediate and short-term effects of eccentric muscle contractions on structural, morphological, mechanical, functional and physiological properties of peripheral nerves: a protocol for a systematic review and meta-analysis”

The aim of the study is to summarize in a systematic review the effects of eccentric exercise on the on structural, morphological, mechanical, functional and physiological properties of peripheral nerves. The study is well designed and the methods proposed are appropriate to answer the research question.

I have few comments and questions that aim to ameliorate the proposed study protocol.

Introduction

The rationale brought by the authors sustain the necessity of conduction this systematic review, however I would be interesting to be more clear about what it represents to muscle function and force production disruption of peripheral nerves following eccentric contraction. Does it relate to the observed decrease in force due to exercise induced muscle damage?

For disruption of peripheral nerves to occur, is it necessary to perform specific eccentric contraction (like isokinetic contractions) or does it also occur when performing functional daily living movements or strength exercises?

I might have missed it, but are these alteration transient? How many days (or hours) does it last? When observed, is the person more susceptible to injury?

As described by the authors, it seems that the peripheral nerve alterations observed after eccentric exercise are somewhat similar those of exercise induced muscle damage. If this is the case, are both things liked? And if so, I would suggest to bring it up.

These comments are made thinking that this is a topic that is not widely discussed so many readers might be reading it for the first time. Addressing possible functional alterations as well as wider possibilities to induce peripheral disruption might strengthen the rationale to and the necessity to study this topic.

2.1.1 Type of studies and 2.1.2 Types of participants

Does the inclusion of studies “evaluating the impact of eccentric contractions (applied to upper or lower limb muscles) on peripheral nerves dysfunction…”, isn’t it possible that the inclusion of individual with peripheral nerves dysfunction will interfere in the results when compared to health tissues? Also, in the following topic “2.1.2 Types of participants” it is specified that will be included studies with “healthy asymptomatic individuals”, isn’t it contradictory? I might be missing a point here.

Why are the following outcomes included:? 4. Physiology: intraneural blood flow measured with Doppler US (B-mode with Colour Doppler)… alterations in signal intensity (T1 and T2 via MRI) and echo intensity (via high-resolution US)

To describe how the strength of the body of evidence, do the authors plane to use the Grading of Recommendations Assessment, Development and Evaluation (GRADE)?

Reviewer #2: The topic of this protocol for SR and MA is novel, unexplored and could potentially summarise an interesting area very unexplored.

I have minor comments, that I hope can improve the protocol.

Abstract. Please specify that eccentric contraction not always cause muscle damage.

Please state clearly that acute reponses from experimental studies will be explored also.

Background:

second paragraph. .. Accordingly, delayed .... please state if this information is at rest state or during exercise.

Page 5. Please clarify what information comes from animal and human studies.

Page 7: for stiffness measure. is it muscle or nerve stiffness??

Page 17: Data synthesis. Second sentence: Meta analysis in animal and human results will be analysed differently or pooled??

Discusion

Can the authors state expected results from preliminary search?

7. PLOS authors have the option to publish the peer review history of their article (what does this mean?). If published, this will include your full peer review and any attached files.

Reviewer #1: No

Reviewer #2: No

---

## [Author Response · Author response to Decision Letter 0]

18 Jun 2023

Letter to Academic Editor

Dear Professor Charlie M. Waugh, 

Thank you for giving us the opportunity to submit a revised draft of the manuscript PONE-D-22-32928. We appreciate the time and effort that you and the reviewers dedicated to providing feedback on our manuscript. We are grateful for the insightful comments and constructive suggestions which help to improve the quality of this manuscript.

We have carefully reviewed the comments and have revised the manuscript accordingly. Those changes are highlighted within the manuscript in yellow. Please see below, for a point-by-point response to the reviewers’ comments and concerns. All page and line numbers refer to the revised manuscript file.

We hope the revised version is now suitable for publication and look forward to hearing from you in due time regarding our submission and to respond to any further questions and comments you may have. 

Furthermore, we would like to clarify that the research center CIPER – Centro Interdisciplinar para o Estudo da Performance Humana (UID/00447/2020), supported by Fundação para a Ciência e a Tecnologia and hosted by Faculdade de Motricidade Humana – Universidade de Lisboa, provides financial support to Raúl Alexandre Nunes da Silva Oliveira, regarding the payment of the publication fee of the manuscript “Immediate and short-term effects of eccentric muscle contractions on structural, morphological, mechanical, functional and physiological properties of peripheral nerves: a protocol for a systematic review and meta-analysis”, for the journal PLOS One. This financial support is just for the publication. None of the authors received a salary from any funder. Raúl Oliveira was supported by Fundação para a Ciência e a Tecnologia, under grant number UID/00477/2020 awarded to CIPER - Centro Interdisciplinar para o Estudo da Performance Humana (unit 447). The author Raúl Oliveira contributed on the preparation of the manuscript. 

Moreover, we declare that we have no competing interests. 

Thank you in advance for your attention and for your consideration. I look forward to hearing from you soon.

Sincerely, 

Dorina Lungu 

Response to Reviewers

We thank both reviewers for their constructive comments, recommendations, and positive insights. We are confident that the comments have helped to improve the manuscript quality. Please consider below our responses for each of the reviewer’s comments, point by point; as well as the changes in the manuscript outlined in yellow accordingly. 

Comments from Reviewer 1 

Comment 1: The rationale brought by the authors sustain the necessity of conduction this systematic review, however I would be interesting to be more clear about what it represents to muscle function and force production disruption of peripheral nerves following eccentric contraction. Does it relate to the observed decrease in force due to exercise induced muscle damage? 

Response: We have followed your recommendation. Please consider the changes in lines #68-69; 83-85 and 89-92. The revised text reads as follows:

“These changes could potentially explain the loss of strength observed after eccentric or unaccustomed exercise.”

“Also, temporary strength loss was present which gradually recovered over time; suggesting that a potential link may exist between the effects of nerve properties and the strength loss..”

“These changes were accompanied by progressive decrease in plantar flexors torque across the bouts. Therefore, consecutive bouts of eccentric contractions with fast angular velocity cause severe nerve properties alterations, and thus impairing skeletal muscle function.“ 

Comment 2: For disruption of peripheral nerves to occur, is it necessary to perform specific eccentric contraction (like isokinetic contractions) or does it also occur when performing functional daily living movements or strength exercises?

Response: It is not to be expected that, in healthy people, nerve damage will occur, and the properties of the nerves will change as a result of performing functional daily living activities, unless a considerable muscular contraction intensity and volume (i.e., repetitions) are associated with those activities.

Comment 3: I might have missed it, but are these alteration transient? How many days (or hours) does it last? When observed, is the person more susceptible to injury?

Response: From our preliminary search, we assume these alterations are transient and there is a potential time course difference between animal and human studies. However, after the validation of the protocol, we hope to obtain answers and conclusions about the question raised. Nevertheless, we added information to the cited studies about the time course of the nerve effects. Please consider the changes in lines # 80-83 and 92-94. The revised text reads as follows:

“The damage to the myelin sheath caused transient sciatic nerve dysfunction, manifested by a significant decline (i.e., -21%) in nerve conduction velocity (NCV) observed on day 7 after eccentric contractions, which returned to baseline values on day 10.14“

“Consistent with these animal studies findings, it has been shown that eccentric contractions induce a temporary reduction on nerve function in healthy individuals.16-19”

Comment 4: As described by the authors, it seems that the peripheral nerve alterations observed after eccentric exercise are somewhat similar those of exercise induced muscle damage. If this is the case, are both things liked? And if so, I would suggest to bring it up.

These comments are made thinking that this is a topic that is not widely discussed so many readers might be reading it for the first time. Addressing possible functional alterations as well as wider possibilities to induce peripheral disruption might strengthen the rationale to and the necessity to study this topic. 

Response: Please note that eccentric exercise with sufficient dose (i.e. both intensity and volume) is suggestive of inducing muscle damage. We assume that, beside the muscle, the nerve is also affected. In the first paragraph we clarify this potential link; and, following your suggestion, we have now improved the introduction to transmit this rationale. 

Comment 5: 2.1.1 Type of studies and 2.1.2 Types of participants

Does the inclusion of studies “evaluating the impact of eccentric contractions (applied to upper or lower limb muscles) on peripheral nerves dysfunction…”, isn’t it possible that the inclusion of individual with peripheral nerves dysfunction will interfere in the results when compared to health tissues? Also, in the following topic “2.1.2 Types of participants” it is specified that will be included studies with “healthy asymptomatic individuals”, isn’t it contradictory? I might be missing a point here. 

Response: Thank you for highlighting this inconsistency. When describing the type of studies, where we wrote “studies evaluating the impact of eccentric contractions (applied to upper or lower limb muscles) on peripheral nerves dysfunction”, we meant to say that studies would be included if they evaluated alterations in different peripheral nerve properties. Thus, “dysfunction” was used to mean the different nerve damage markers that we described afterwards. However, we recognize that it may induce some bias in the way the information is interpreted. Please consider the changes in lines # 121-124, as the revised text reads as follows:

“The review will include randomised (RCTs) and non-randomised controlled trials or quasi-experimental studies evaluating the effects of eccentric contractions (applied to upper or lower limb muscles) on peripheral nerves structural, morphological, mechanical, functional and physiological properties.” 

Comment 6: Why are the following outcomes included:? 4. Physiology: intraneural blood flow measured with Doppler US (B-mode with Colour Doppler)… alterations in signal intensity (T1 and T2 via MRI) and echo intensity (via high-resolution US)

Response: We have decided to include these outcomes because changes at these levels could be associated with alterations in peripheral nerve physiology and could be considered as nerve damage markers. For instance, increased intraneural blood flow could be attributed to neuronal inflammation that results in activated inflammatory cytokines, induction of nitric oxide synthase and increased vasodilation. Increased intraneural blood flow and changes in signal intensity (T1 and T2 via MRI) and echo intensity (via high-resolution US) were observed in clinical population. Please consider below some references to support the aforementioned justification.

References:

o Gao, Y., Weng, C., & Wang, X. (2013). Changes in nerve microcirculation following peripheral nerve compression. Neural regeneration research, 8(11), 1041–1047. https://doi.org/10.3969/j.issn.1673-5374.2013.11.010

o Vanderschueren, G. A., Meys, V. E., & Beekman, R. (2014). Doppler sonography for the diagnosis of carpal tunnel syndrome: a critical review. Muscle & nerve, 50(2), 159–163. https://doi.org/10.1002/mus.24241

o Boaventura, M., Sastre-Garriga, J., Garcia-Vidal, A., Vidal-Jordana, A., Quartana, D., Carvajal, R., Auger, C., Alberich, M., Tintoré, M., Rovira, À., Montalban, X., & Pareto, D. (2022). T1/T2-weighted ratio in multiple sclerosis: A longitudinal study with clinical associations. NeuroImage. Clinical, 34, 102967. https://doi.org/10.1016/j.nicl.2022.102967

o Sollmann, N., Weidlich, D., Klupp, E., Cervantes, B., Ganter, C., Zimmer, C., Rummeny, E. J., Baum, T., Kirschke, J. S., & Karampinos, D. C. (2020). T2 mapping of the distal sciatic nerve in healthy subjects and patients suffering from lumbar disc herniation with nerve compression. Magma (New York, N.Y.), 33(5), 713–724. https://doi.org/10.1007/s10334-020-00832-w

o Fan, C., Fede, C., Pirri, C., Guidolin, D., Biz, C., Macchi, V., De Caro, R., & Stecco, C. (2020). Quantitative Evaluation of the Echo Intensity of Paraneural Area and Myofascial Structure around Median Nerve in Carpal Tunnel Syndrome. Diagnostics (Basel, Switzerland), 10(11), 914. https://doi.org/10.3390/diagnostics10110914

Comment 7: To describe how the strength of the body of evidence, do the authors plane to use the Grading of Recommendations Assessment, Development and Evaluation (GRADE)? 

Response: Thank you for this suggestion. It would have been interesting to explore this aspect. However, we did not consider the application of the Grading of Recommendations Assessment, Development and Evaluation (GRADE), since the main goal of this article is to reveal the effects of eccentric contractions on peripheral nerve properties instead of providing recommendations in favour or against the intervention. 

 

Comments from Reviewer 2 

Comment 1: Abstract. Please specify that eccentric contraction not always cause muscle damage.

Response: Lines 24-25: “It is widely acknowledged that eccentric muscle contractions may cause skeletal muscle damage.”

We used the expression “may cause skeletal muscle damage” on purpose to point out that it doesn't always happen but depending on the characteristics of the eccentric contractions it can occur. In the introduction, lines 56-59, we specify in which conditions the muscle damage can occur. 

“There is a considerable body of literature showing that eccentric contractions may induce signs and symptoms of muscular damage2; in particular to unaccustomed eccentric contractions3, performed at high intensity4 and with fast muscle-tendon elongation velocity.5”

Comment 2: Please state clearly that acute reponses from experimental studies will be explored also.

Response: Only the acute responses will be examined, and we refer to them as “immediate and short-term effects”. Please consider the lines 27-30 and 106-110. 

Abstract: “The purpose of this review is to examine whether eccentric muscle contractions induce immediate and short-term effects on structural, morphological, mechanical, functional and physiological properties of peripheral nerves, from both animal and human studies. “ 

Introduction: “To this end, this study aims to systematically review the literature in order to examine the immediate (i.e., <2 hours) and short-term (i.e., <10 days) effects of eccentric muscle contractions on peripheral nerve structural, morphological, mechanical, functional and physiological properties, from both animal and human studies.”

Comment 3: Background: second paragraph. .. Accordingly, delayed .... please state if this information is at rest state or during exercise.

Response: Thank you for the suggestion. We have included this information. Please consider the changes in lines # 70-72. The revised text reads as follows:

“Accordingly, delayed sensory and motor conduction velocity of the radial nerve was observed at rest in the dominant arms of tennis players compared with their non-dominant arms and with control individuals.12”

Comment 4: Page 5. Please clarify what information comes from animal and human studies.

Response: Thank you for the suggestion. We have followed your recommendation. Please consider the changes in lines # 79-80 and 92-94. The revised text reads as follows:

“Thus, animal studies revealed myelin sheath damage, observed after 20 eccentric contractions of the plantar flexors with fast angular velocity (180°/s).14”

“Consistent with these animal studies findings, it has been shown that eccentric contractions induce a temporary reduction on nerve function in healthy individuals.16-19”

Comment 5: Page 7: for stiffness measure. is it muscle or nerve stiffness??

Response: All the outcomes are relative to the peripheral nerve. We have, accordingly, modified the first paragraph of 2.1.4 Types of outcome measures, to make it more explicit and emphasize this aspect. Please consider the changes in lines # 138-140. The revised text reads as follows:

“We will evaluate the immediate (i.e., up to 2 hours after eccentric contraction) and short term (i.e., up to 10 days) effects of eccentric contractions on structural, morphological, mechanical, functional and physiological properties of the peripheral nerves. “

Comment 6: Page 17: Data synthesis. Second sentence: Meta analysis in animal and human results will be analysed differently or pooled??

Response: Thank you for the comment. We will analyse separately the results from animal and human studies. Please consider the changes in lines # 228-230. The revised text reads as follows:

“Meta-analyses for the animal and human studies will be performed separately and applied for each outcome of interest of nerve properties.”

Comment 7: Discusion. Can the authors state expected results from preliminary search?

Response: Thank you for pointing this out. We have included a short sentence that addresses the expected results. Please consider the changes in lines # 239-241. The revised text reads as follows:

“From the preliminary search, alterations in some outcomes of interest under study are expected, such as decreased nerve function, compatible with peripheral nerve damage, both from animals and human studies.”

---

## [Editor Report · Decision Letter 1]

19 Jul 2023

Immediate and short-term effects of eccentric muscle contractions on structural, morphological, mechanical, functional and physiological properties of peripheral nerves: a protocol for a systematic review and meta-analysis

PONE-D-22-32928R1

Dear Dr. Lungu,

We’re pleased to inform you that your manuscript has been judged scientifically suitable for publication and will be formally accepted for publication once it meets all outstanding technical requirements.

Kind regards,

Charlie M. Waugh

Academic Editor

PLOS ONE

---

## [Editor Report · Acceptance letter]

21 Jul 2023

PONE-D-22-32928R1 

Immediate and short-term effects of eccentric muscle contractions on structural, morphological, mechanical, functional and physiological properties of peripheral nerves: a protocol for a systematic review and meta-analysis 

Dear Dr. Lungu:

I'm pleased to inform you that your manuscript has been deemed suitable for publication in PLOS ONE. Congratulations! Your manuscript is now with our production department. 

Kind regards, 

on behalf of

Dr. Charlie M. Waugh 

Academic Editor

PLOS ONE